# Retinal photoisomerization versus counterion protonation in light and dark-adapted bacteriorhodopsin and its primary photoproduct

Partha Malakar[1,5], Samira Gholami[2,5], Mohammad Aarabi [2], Ivan Rivalta [2,3], Mordechai Sheves [4] ✉, Marco Garavelli [2] ✉ & Sanford Ruhman[1] ✉

Discovered over 50 years ago, bacteriorhodopsin is the first recognized and most widely studied microbial retinal protein. Serving as a light-activated proton pump, it represents the archetypal ion-pumping system. Here we compare the photochemical dynamics of bacteriorhodopsin light and dark-adapted forms with that of the first metastable photocycle intermediate known as "K". We observe that following thermal double isomerization of retinal in the dark from bio-active all-*trans 15-anti* to 13-*cis, 15-syn*, photochemistry proceeds even faster than the ~0.5 ps decay of the former, exhibiting ballistic wave packet curve crossing to the ground state. In contrast, photo-excitation of K containing a 13-*cis, 15-anti* chromophore leads to markedly multi-exponential excited state decay including much slower stages. QM/MM calculations, aimed to interpret these results, highlight the crucial role of protonation, showing that the classic quadrupole counterion model poorly reproduces spectral data and dynamics. Single protonation of ASP212 rectifies discrepancies and predicts triple ground state structural heterogeneity aligning with experimental observations. These findings prompt a reevaluation of counter ion protonation in bacteriorhodopsin and contribute to the broader understanding of its photochemical dynamics.

Discovered over 50 years ago[1], bacteriorhodopsin (BR) is the first recognized microbial retinal protein (MRP). It is a light-activated proton pump contained in the purple membrane of *Halobacterium salinarum* and in view of extensive investigation of its inner workings, serves as an archetypal ion-pumping protein. Upon photoexcitation, the all-*trans, 15-anti* retinal chromophore (AT) bound to LYS216 via a protonated Schiff base (PSB) linkage, isomerizes in ~0.5 ps to the first metastable intermediate 13-*cis, 15-anti* configuration (see Fig. 1), whose absorption peaks at 590 nm ($K_{590}$, or K). This triggers a 15 msec cyclic

series of structural changes resulting in proton translocation from the cytoplasmic to the extracellular domain. Since BR's discovery many more microbial retinal proteins (MRPs) have been identified, and study of additional proteins reveals a broad variation of photo-isomerization dynamics[2-12]. Efforts to correlate these variations with a specific protein's absorption peak, isomerization quantum efficiency, or biological function have been largely frustrated. In the dark, BR equilibrates thermally in a process coined dark adaptation (DA) to a balanced mixture of the biologically active AT resting state

[1]Institute of Chemistry, The Hebrew University of Jerusalem, Jerusalem 9190401, Israel. [2]Dipartimento di Chimica industriale "Toso Montanari", Università di Bologna, Viale del Risorgimento 4, 40136 Bologna, Italy. [3]ENSL, CNRS, Laboratoire de Chimie UMR 5182, 46 allée d'Italie, 69364 Lyon, France. [4]Department of Molecular Chemistry and Materials Science, The Weizmann Institute of Science, Rehovot 7610001, Israel. [5]These authors contributed equally: Partha Malakar, Samira Gholami. ✉e-mail: mudi.sheves@weizmann.ac.il; marco.garavelli@unibo.it; sandy@mail.huji.ac.il

**Fig. 1 | The chromophoric species in BR. A** Different initial resting states of the same microbial retinal protein (MRP), i.e. bacteriorhodopsin (BR), including dark-adapted (DA, in gray), light-adapted (LA, in blue) and $K_{590}$ photo-intermediate (K, in red). The gray cube and the shading rectangles show the membrane and embedded chromophoric species inside the active site of corresponding protein, respectively. **B** The corresponding chromophoric species.

and another with doubly isomerized *13-cis, 15-syn* (13C) retinal chromophore. The latter fully reverts to AT over repeated photo-excitation (light-adaptation, LA), as shown in Fig. 1[13]. Preliminary comparison showed that photoisomerization of 13C proceeds much faster than AT, exhibiting continuous excited state spectral evolution assigned to ballistic wave packet curve crossing to the ground state[14]. On the other hand, in ballistic photoisomerization, retinal motion after photoexcitation is relatively unaffected by external forces or interactions. In this state, the retinal undergoes efficient isomerization and returns to its ground state without significant interference from its surroundings. This behavior is often associated with faster and more efficient photoisomerization.

The $C_{13}$-N portion of the retinal chromophore and its nearby environment differs most between these three ground-state[15,16] variants and is involved in retinal deprotonation. In particular, local water molecules as well as the protonation state of nearby titratable residues such as ASP85, ASP212, and ARG82 (the so called *complex counterion*) is of paramount importance for setting the right local electrostatic environment and pKa gradients promoting retinal deprotonation and the following proton-pumping events. Quantum chemical simulations on both model systems and retinal proteins have indeed suggested that local electrostatics are crucial also in tuning the peak absorption and effecting the rate and quantum efficiency of the photoisomerization by modulating the energy gaps between the $S_0$, $S_1$ and $S_2$ states and molding topology of the isomerization reaction coordinate[17–20]. The chromophore together with its complex counterion system are widely believed to form an overall neutral quadrupole[21], associated with two positive (the retinal PSB and the protonated arginine ARG82) and two negative (the deprotonated aspartates ASP85 and ASP212) charges[22–24].

In this work, we aim to complete this effort by recording pump-probe measurements on yet another ground state BR reactant, the first photo-intermediate K, and comparing the results from all three (AT, 13C, K) with QM/MM calculations (see Method section, Supplementary note 1 and Supplementary Fig. 1 for details). We assume an identical protein environment makes for a more meaningful

comparison, and improved testing for the predictive capacity of theoretical models[14,25,26]. We focus attention on the $C_{13}$-N portion of the retinal chromophore. Accordingly, we employ QM/MM calculations to compare the accepted quadrupole model with an alternative scenario where one aspartate (ASP212) is protonated and the system is overall positively charged. Prompted by a systematic error of calculated photoisomerization properties and ground state absorption spectra of the tested reactant states using the conventional counter ion model, we show that this alternative protonation state does significantly better in reproducing spectroscopic data and the strong resting state dependence of excited state potential barriers to isomerization, possibly explaining the ballistic *vs* non-ballistic isomerization dynamics of the 13C and AT respectively, and slower and markedly multi-exponential excited state decay recorded for K. Simulations suggest the latter reflects coexistence of different conformations for this new protomeric state, unlike LA and DA states that are dominated by a unique conformation. This is consistent with ASP85 being deproto-nated and acting as the primary proton acceptor, and questions the validity of an electroneutral retinal binding pocket in BR. Eventually, the goal of this study is to understand the factors influencing retinal spectral tuning and photoisomerization in BR and beyond. Finally, we show that the fraction of BR pigments transformed to 13C in the dark is two times less than previous estimates[27–32]. This is biologically significant since it ensures a large majority of pigments are ready to function upon irradiation even after prolonged darkness, and do not require the inefficient (~3%) light adaptation process to convert non-productive *cis* into all-*trans*[33,34].

## Results

### Photochemistry in dark- and light-adapted bacteriorhodopsin

An earlier study from our laboratory found that 13C photochemistry is much faster than AT, and that at late pump-probe delays, excited state's transient absorption (TA) bands were exclusively due to the AT reactant[14]. TA of LA- and DA-BR were recorded here anew following sample excitation near the isosbestic point of DA and LA absorption (Fig. 2A and Supplementary Fig. 2) to quantify the isomer ratios in our

DA samples, and to probe the stimulated emission band all the way to the NIR. These were necessary for determining the quantum efficiency of photo-isomerization in both 13C-BR and AT-BR. Isosbestic pumping ensures equal photoexcitation probability for both isomers in a DA sample, facilitating separation of their contributions to the signal. The broad TA spectral coverage from 430 to 1400 nm used in these experiments allows full resolution of excited-state absorption (ESA) and stimulated emission (SE) transitions from $S_1$, together with ground state bleaching (GSB).

As depicted in Fig. 2B, the fluorescent state of AT bacteriorhodopsin in LA samples exhibits an ~100 fs phase of complex spectral evolution, followed by curve crossing to $S_0$ with a characteristic decay time of ~500 fs (Fig. 2D)[15,16]. This is accompanied with bifurcation to the initial AT resting state or the isomerized photoproduct with roughly equal probability[35]. The latter retinal chromophore is in a vibrationally hot 13-*cis*–15-*anti* configuration, which cools over a few picoseconds to the microsecond lived intermediate denoted as K. TA data of DA-BR is presented as a two-dimensional color-map in Supplementary Fig. 3b in the Supplementary Information revealing the presence of a fast decay component relative to changes accompanying photo-isomerization of LA-BR.

Having excited at the isosbestic point between LA and DA spectra, pure 13C can be obtained by subtracting that due to AT reactants using previously determined values for isomer ratios in DA-BR[36]. Surprisingly, subtraction based on literature values for this ratio leaves an excess AT contribution in the DA data (see Supplementary Information for details). Assuming independence of each isomer contribution to the TA spectra of the mixed DA sample, dynamic difference spectra ($\Delta\Delta OD = \Delta OD(t + \Delta t) - \Delta OD(t)$) are employed to determine their correct ratio in the dark. While early finite-difference spectra vary for LA

and DA-BR throughout the probed spectral range (see Supplementary Fig. 4a in the Supplementary Information), they converge for $t \geq 600$ fs once AT data is multiplied by a constant factor of $0.65 \pm 0.03$, in spectral regions dominated by the excited state (see Supplementary Fig. 4c, d in the Supplementary Information). As described in the Supplementary Information, further measures were taken to verify this ratio, including tuning the excitation wavelength and following NIR stimulated emission signal amplitudes that are exclusively measures of $S_1$ population. All of these tests uphold this unusual ratio indicating that AT-BR remains the dominant species even after DA.

Thus, extracted 13C transient absorption data is presented as a 2-D colourmap in Fig. 2C. The TA spectrum at early pump-probe delays is similar in appearance to that for AT (see Fig. 2B) but differs in temporal evolution in two ways: (1) difference bands due to the excited state dissipate much faster and are erased within ~300 fs (Fig. 2D) and (2) this disappearance is accompanied by continuous excited state absorption and emission shifts to the blue and to red, respectively - hallmarks of ballistic photoisomerization[37]. The latter is observed here via extending the probing window into the NIR region. This ultrafast photoisomerization leaves behind a difference spectrum that, unlike AT, is almost entirely positive, as expected for isomerization from a *cis* to AT configuration. This positive difference dipole strength serves, here, for estimating quantum efficiency for isomerization of the 13C pigment. A comparison of absorption dipole strengths of several 13C and AT retinal pigments show a systematic increase of ~15% of the absorption dipole strengths to $S_1$ between them[38,39]. Assuming this holds for photoisomerization of 13C to AT-BR as well, the band integral of the observed difference spectrum at long delays predicts a large quantum efficiency of more than 50% (Detailed in Supplementary Information). Accordingly, the extremely fast photochemistry of the

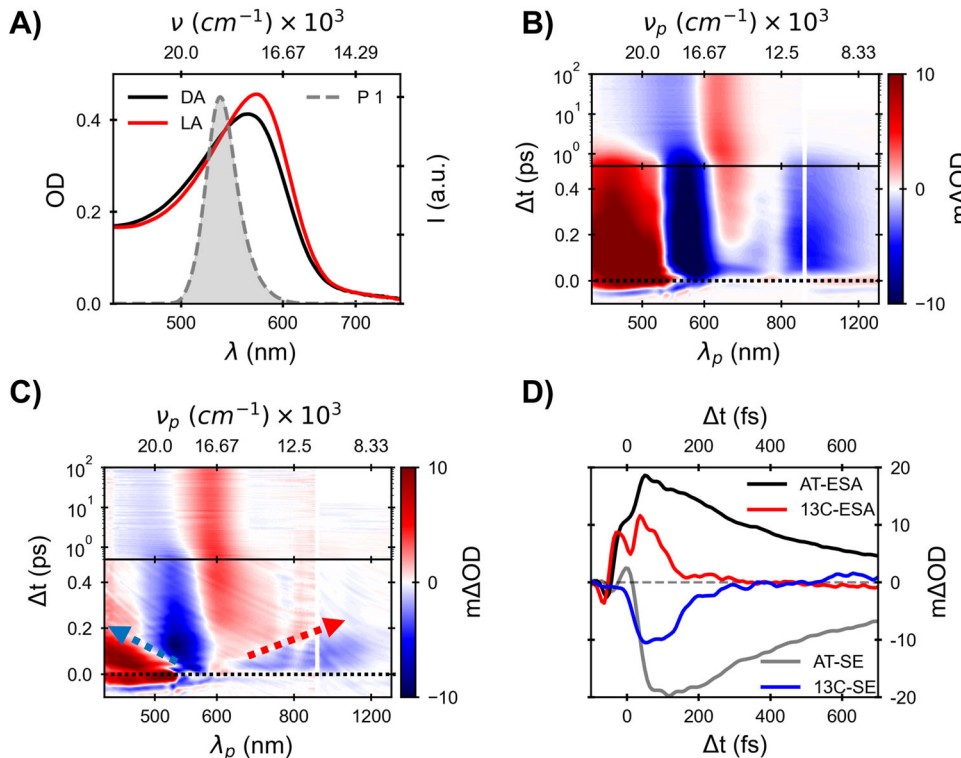

**Fig. 2 | Spectroscopic Characterization of light-adapted and dark-adapted Bacteriorhodopsin Photoisomerization. A** Absorption spectra of light-adapted bacteriorhodopsin (LA-BR) and dark-adapted bacteriorhodopsin (DA-BR) line plot with excitation pulse spectrum in area plot. The 2D colourmap of TA experimental data of **B** all-*trans* (AT) or LA-BR and **C** 13-*cis* bacteriorhodopsin (13C-BR). In the 2D color maps, the X-axis represents probe wavelength (or wavenumber) while the

Y-axis represents the delay between pump and probe. Initial 500 fs pump-probe delay is linear, and the rest of the axis is logarithmic. The absorption difference is color-coded: the color bar at the right of the map presents the value of absorption difference. **D** Excited-state absorption (ESA, 470 nm) and stimulated emission (SE, 930 nm) decay of AT-BR and 13C-BR (SE decay is multiplied with 4). Source data are provided as a Source Data file.

*cis* resting state is not due to curve crossing to $S_0$ very near the Frank Condon state in terms of structure, since that would also lead to minor isomerization efficiency. Furthermore, the estimated QY proves that the ~1/30 yield of light adaptation is determined by later phases of the *cis* photo-cycle[33–35].

Three dominant factors have been proposed to control excited state decay dynamics of PSBs in various environments. One is the potential energy surface (PES) topography along the torsional motion that brings to the $S_1/S_0$ conical intersections (CI). Another involves the $S_1/S_2$ energy gap and ionic /covalent-character mixing along the C=C bond relaxation coordinate (which is often referred as the bond length alternating, BLA, coordinate), and a third is the energy drop associated to the same C=C bond relaxation from the Franck-Condon (FC) region[15,40]. Here, in order to identify the origin of the different ballistic *vs* non-ballistic dynamics observed for 13C (in DA-BR) *vs* AT (in DA- and LA-BR), we computed the QM(CASPT2//CASSCF)/MM photoisomerization minimum energy paths (MEPs) from the FC to CIs of 13C in DA-BR and AT in LA-BR adopting different PSB/complex counterion models (see Supplementary Fig. 8 and Fig. 3).

The results for the standard counterion model (displaying fully deprotonated ASP85 and ASP212 and thus holding an overall neutral charge, see Fig. 4A) are shown in Supplementary Fig. 8 of

the Supplementary Information. Surprisingly, 13C and AT show high similarities in their excited states pathways, despite the different behavior observed in experiment. Interestingly, a very strong covalent/ ionic mixing, due to $S_1$ and $S_2$ being very close in energy, is observed along the $S_1$ MEP that describes the C=C bond relaxation pathway out of the FC region, eventually leading to a stable $S_1$ minimum with even C−C bond lengths (EBL) and a significant covalent character. Thus, in order to eventually proceed with the $C_{13}=C_{14}$ photoisomerization, extra energy is required to photoisomerize along a $C_{13}=C_{14}$ bond that is not elongated enough to feature a barrierless torsion. More remarkably, a larger barrier is predicted for 13C (4.4 kcal mol$^{-1}$) as compared to AT (3.1 kcal mol$^{-1}$), which should eventually result into its slower photoisomerization at odd with observations.

These results are confirmed by semi-classical QM/MM excited states dynamics of both 13C and AT, as reported in the Supplementary Information (see Supplementary Fig. 9), highlighting a long living excited state population that is trapped in the EBL minimum. While this picture has been shown to account for the slower dynamics observed for retinal PSB in methanol solution[15], it appears incompatible with the ballistic (and therefore substantially barrierless) nature of the excited state decay in 13C, as well as with the sub-ps excited state dynamics observed for AT.

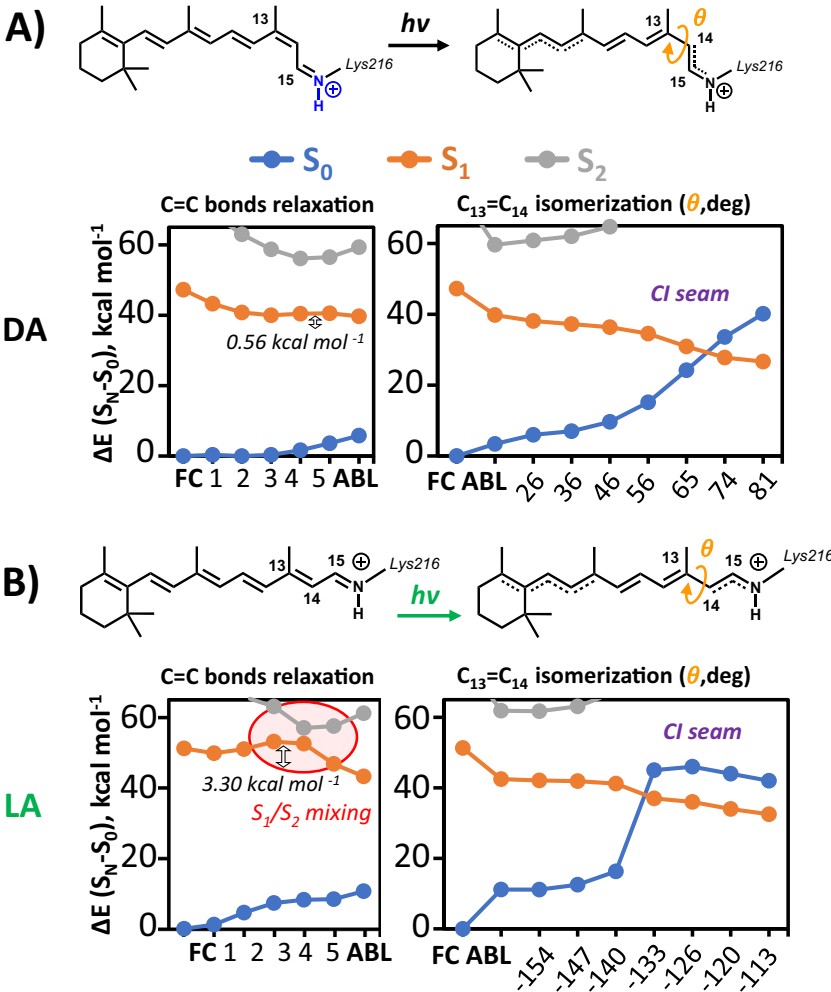

**Fig. 3 | The modeling of 13-*cis* (13C) and all-*trans* (AT) photochemistry.** CASPT2// CASSCF/MM prediction of minimum energy paths (MEPs) for C=C bond relaxation (defined by the interpolation from the Franck-Condon, FC, to the alternate bond-lengths, ABL, structures, as depicted in the sketch) and for $C_{13}=C_{14}$ photo-isomerization (defined by the interpolation of torsional angle $\theta$ towards the CI seam) of **A** 13C in dark-adapted bacteriorhodopsin (DA-BR) and **B** AT in light-adapted bacteriorhodopsin (LA-BR). The values 0.56 and 3.30 kcal mol$^{-1}$ are associated with the energy barriers along the C=C bond relaxation path of DA and LA, respectively. The orange shaded area in the panel B indicates the $S_1/S_2$ mixing region along the C=C bond relaxation photoreaction coordinate. Source data are provided as a Source Data file.

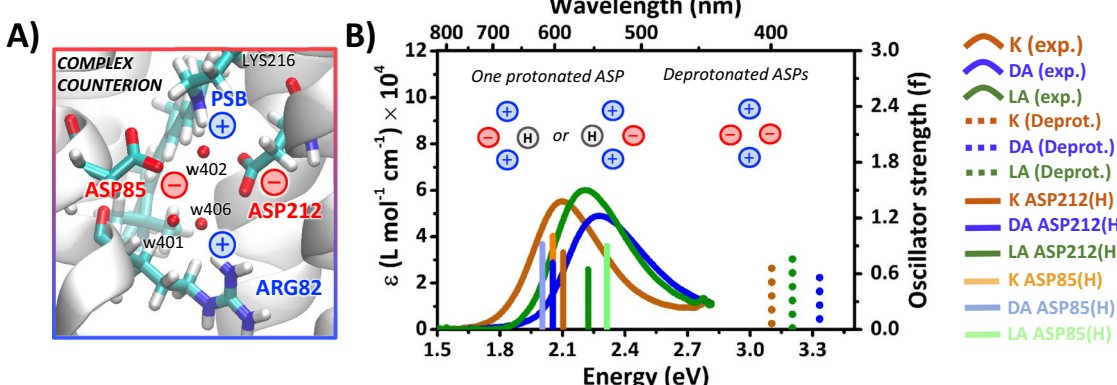

**Fig. 4 | Linear absorption of different resting states of BR and counterion models. A** The retinal chromophore bound to LYS216 via a positively charged PSB is surrounded by an extended H-bonding network involving another positive sidechain, ARG82, three water molecules, i.e. w401, w402 and w406, and two protonable aspartates, ASP85 and ASP212. **B** Experimental absorption spectra of the three different resting states (solid curves) are compared with QM/MM theoretical predictions of electronic ($S_0 \rightarrow S_1$) transitions energies (at CASPT2// CASSCF/MM level), using the standard complex counterion model with two deprotonated aspartates (dashed sticks) or one protonated ASP sidechain (solid sticks), labeled as "(H)". The "+" and "−" signs refer to the positive and negative charges associated with the different residues, respectively. Source data are provided as a Source Data file.

This inconsistency prompted us to identify any flaw in our reactive model. Accordingly, reaction coordinate profiles were calculated for different counterion models where one of the two aspartates (either ASP85 or ASP212) is protonated, and thus holding an overall positive charge. For sake of clarity, here we discuss only the model with protonated ASP212, as it is the only one delivering experimentally consistent results and given the fact that ASP85 is established to act as the primary proton acceptor from PSB in the BR photocycle[41-44], thus making its protonation unrealistic. Figure 3 shows the QM/MM results of this model for 13C (found only in DA) and AT (found both in LA and DA). More specifically, both the C=C bond relaxation and the $C_{13}=C_{14}$ photoisomerization pathway (highlighting the two consecutive steps of the relaxation dynamics following excitation) have been mapped. Upon a similar energy drop leaving the FC structure, the computed C=C bond relaxation profiles of 13C and AT are significantly different. Unlike the MEP calculated for 13C using the standard counterion model, here it is almost barrierless (with an energy barrier <1 kcal mol⁻¹). In contrast for LA-BR, the energy profile of AT still features a local minimum (with an EBL structure) as in the standard model. However, while there the EBL structure was the only stable planar minimum found on $S_1$, a ca. 3 kcal mol⁻¹ energy barrier is involved here in reaching a much more stable planar $S_1$ minimum with alternated bond lengths (ABL structure), a fully elongated $C_{13}=C_{14}$ bond and a strong ionic character. From this point, the $C_{13}=C_{14}$ photoisomerization pathways of both systems show a barrierless energy profile, leading to a peaked CI for 13C and to an extended CI seam for AT. This scenario indicates that, within the ASP212 protonated model, the C=C bond relaxation represents the rate determining step of the retinal photoisomerization for both 13C and AT and that the elongation of $C_{13}=C_{14}$ towards the ABL structure, and its photoisomerization, is predicted to be faster in 13C than in AT, in agreement with experimental observation. These conclusions are supported by semi-classical QM/MM excited states dynamics of both 13C and AT: as reported in the Supplementary Information (see Supplementary Fig. 9), a ballistic decay is predicted for 13C *vs* longer decay dynamics for the AT.

To further test the appropriateness of the unconventional counterion protonation model, computed energy gaps were compared with observed $S_0 \rightarrow S_1$ transition energies (i.e., vertical excitations, VE), as depicted in Fig. 4B. The LA and DA forms of BR are characterized by 570 nm and 560 nm absorption maxima ($\lambda_{max}$), the former composed exclusively of AT, and the latter a mixture of AT and 13C retinal chromophores[36]. Knowing the isomer composition in DA state allows extraction of the absorbance of 13C as well. Using the 0.35:0.65 ratio

for the 13C to AT isomers in the dark determined here, the 13C absorption spectrum included in Fig. 4B peaks at 543 nm. Interestingly this $\lambda_{max}$ value is in much better agreement with that predicted for 13C-BR (539 nm) from the well-known relationship between absorption energies and C=C stretching frequencies of MRPs than estimates based on isomer ratios of ~1[45,46].

Multireference/multiconfigurational QM approaches, named CASPT2//CASSCF, in conjunction with MM computations have been shown very effective at predicting VEs of retinal proteins in general[37,47,48]. However, as shown in Fig. 4B, while using the standard complex counterion protonation states of aspartates (i.e., both negatively charged), heavily blue shifted (by approximately 1 eV) VEs are predicted for AT, 13C and K. These deviations far exceed the typical error of ~0.2 eV expected for CASPT2//CASSCF/MM methods, which further strength the suggestion that environmental electrostatics are not properly addressed in the standard protonation model. Very notably, as depicted in Fig. 4B, the computed VEs fall within 0.2 eV from the corresponding experimental $\lambda_{max}$ when either ASP212 or ASP85 are protonated. While computed transition energies suggest that one of the two aspartates must be protonated in the ground state of AT, 13C, and K, they do not provide information on which of the two sidechains is protonated, i.e., ASP212 or ASP85.

The standard complex counterion model has been also supported by several infrared (IR) spectroscopy measurements[49,50], involving isotopic labeling and differential spectra between various resting states (including LA and K) or between wild-type and single-site mutants in order to overcome the IR signals congestion in the spectra of single proteic systems. These studies have highlighted two main band regions of the LA system associated to strong H-bonding of the three water molecules bridging the charged sidechains and AT-PSB (see Supplementary Fig. 10 in the Supplementary Information). The first IR band ($\nu_{NH}$) at ~2800 cm⁻¹, assigned to the N−H stretching of the PSB, is associated to a mixed mode of N−H and O−H stretchings from PSB and w402, respectively. The second band involves a mixed mode of N−H and O−H stretchings from ARG82 and w406 ($\nu_1$), respectively, associated to strong H-bonds linking to negatively charged ASP212, at ~3000 cm⁻¹ and the O−H stretching mode of w401 ($\nu_2$), strongly interacting with negatively charged ASP85, at ~3100 cm⁻¹.

In order to assess the reliability of the various (fully deprotonated or singly protonated aspartate) models of LA, we have simulated IR spectra for both protonation states by means of density functional theory (DFT) accounting for a QM description of the entire complex counterion within a QM/MM scheme (see Supplementary Information,

Supplementary Fig. 10), i.e., B3LYP/MM. As shown in the Supplementary Information, calculated IR spectra simulated for a protonated ASP212 is not only consistent with experiment, but actually much more so than the doubly deprotonated (standard) or the protonated ASP85 models supporting the proposed structure. Remarkably, this is consistent with ASP85 acting as the primary acceptor for retinal deprotonation, as widely recognized and consolidated[39–41]. Our conclusion is also supported by the fact that previous study[51] showed that adding anharmonicities and including a larger QM region will finally further red-shift the NH frequency, indicating the discrepancy of the standard counterion model.

### Photochemistry of the K intermediate

The strong dependence of BR photoisomerization to retinal configuration detected in LA and DA samples have prompted us to expand this survey to other ground state initial configurations in BR. $K_{590}$, the first metastable $S_0$ photocycle intermediate is an obvious choice, being sufficiently long lived for characterization of the its photochemistry in isolation. For this purpose, LA-BR is excited with a 100-fs actinic pulse peaking at 570 nm, which relaxes within picoseconds to a mixture of AT and K. 60 ps after the actinic excitation this mixture was excited again with a ~10-fs pulse centered at 625 nm (see Fig. 5 and Supplementary Fig. 11). This delay ensures all actinic photoproducts, those isomerized to K as well as the non-reactive fraction that reforms AT pigments, relax completely in $S_0$. As in the separation of 13C signals from the mixed DA samples, extraction of K photodynamics can then be obtained by subtraction of the signal contribution of AT-BR to the signals.

The TA data following the actinic excitation is presented in Supplementary Fig. 12b and Supplementary Fig. 12d in the Supplementary Information. Following the photoexcitation, TA spectrum of the mixture shows ESA, GSB and SE like pure AT. However, the excited state signatures (ESA and SE) for the mixture extend to much longer delays, suggesting a surprisingly longer lifetime of fluorescent state of K* even relative to that in AT BR. Additionally, the TA spectra for the actinic mixture with K is identical beyond a 50 ps delay with an experiment without actinic photolysis after multiplying by a factor >1 (see Supplementary Fig. 13 in the Supplementary Information), consistent with K* either isomerizing to AT-BR or reforming K.

To quantify the fractional contribution of AT in TA data to be subtracted for isolating K signals, the amplitude of impulsive wave packet modulations was compared for pump-probe with and without an actinic preparation pulse. Photoexcitation with a short pulse creates low frequency coherent wave packet motion in the $S_1$, which produces observable oscillatory spectral modulation in TA at ~520 nm (see Supplementary Fig. 9 and Supplementary Fig. 14 in the Supplementary Information)[52]. The coherent modulation of probe transmission at this wavelength for the actinic mixture appears identical to that obtained after exciting an AT sample after multiplying by factor of 0.6 (Supplementary Fig. 14 in the Supplementary Information). Assuming the oscillatory component is exclusively due to AT response, subtracting AT data multiplied by 0.6, renders our approximation of pure K TA. This factor was independently verified by measuring anisotropy of the TA spectra by varying the relative polarization orientations of pump and probe pulses. Details of these measurements are provided in the Supplementary Information, with results fully supporting validity of subtraction factor of 0.6 as well.

TA data assigned to K are presented in Fig. 6A, B. The spectral structure immediately following photoexcitation exhibit customary ESA to the blue and SE to the red of a GSB, which for K is shifted to the red (590 nm) as expected. The subsequent decay of excited state bands is surprisingly slow and obviously multi-exponential (Fig. 6C). It includes a minor initial decay phase which is limited to ~300 fs, followed by biphasic decay well fit as bi-exponential with equal amplitude 1.7 ps and 11 ps components. The extended nature of K* decay is particularly surprising in view of vibrational spectra of K in $S_0$, indicating a significant pre-twisting of the retinal backbone near $C_{13}=C_{14}$, which has been identified frequently in the literature as predicting very rapid photoisomerization dynamics[53–56]. Our finding that decay of K* is on average much slower than that of I, the fluorescent state of $BR_{570}$ was indirectly suggested from fluorescence quantum yield studies reported by Atkinson and coworkers[57].

To gain more insight into the spectral evolution, we calculated the dynamic difference of the TA spectrum over these effectively time separated decay stages (see Fig. 6D). That for the range of 400–200 fs shows negative and positive differences at 470 and 850 nm, respectively, indicating the disappearance of the fluorescent state. In addition, it exhibits a positive feature in the range 520–700 nm, which is related to the isomerization and bleach recovery. The difference spectra of 400–200 fs and 2000–400 fs differ. The positive peak is more red-shifted for later delay ranges, suggesting the short-lived process might have higher isomerization efficiency. Later dynamic difference spectra extending to 10 ps are all identical in spectral structure (see Fig. 6D), which is compatible with both picosecond

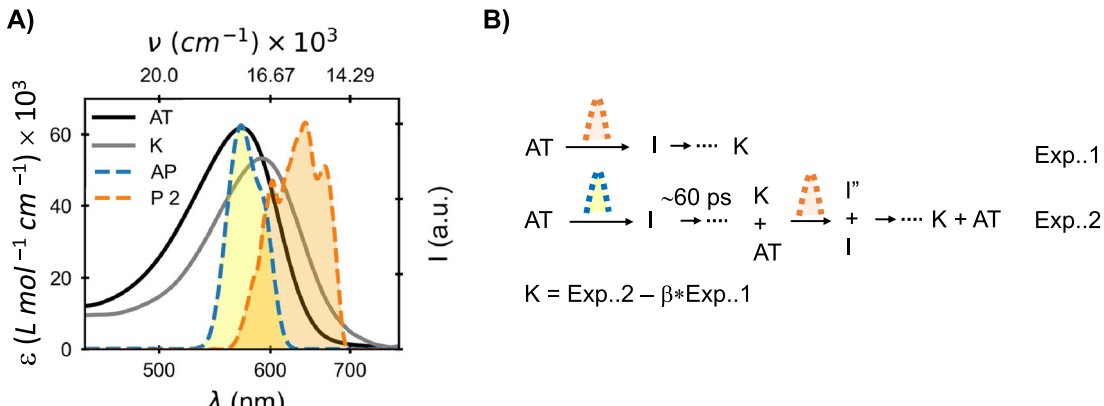

**Fig. 5 | AT and K absorption spectra, excitation spectra and AT photoreaction scheme. A** Absorption spectra of all-*trans* (AT) and K intermediate along with excitation pulse spectra. **B** A representative scheme of AT photoreaction in light-adapted bacteriorhodopsin (LA-BR) and in the presence and absence of K. It also shows the photoreaction of K and isolation of pure K photochemistry. Exp..1 represents the AT-BR's photochemistry. Exp..2 uses two excitation pulses, a few ps after the actinic excitation of AT-BR a mixture of AT-BR and K intermediate was produced. TA measurement after 60 ps of the actinic excitation records the photochemistry of the AT-BR and K. Subtracting the correct weighted contribution of AT-BR's data from Exp..2 gives the photochemistry of K. Source data are provided as a Source Data file.

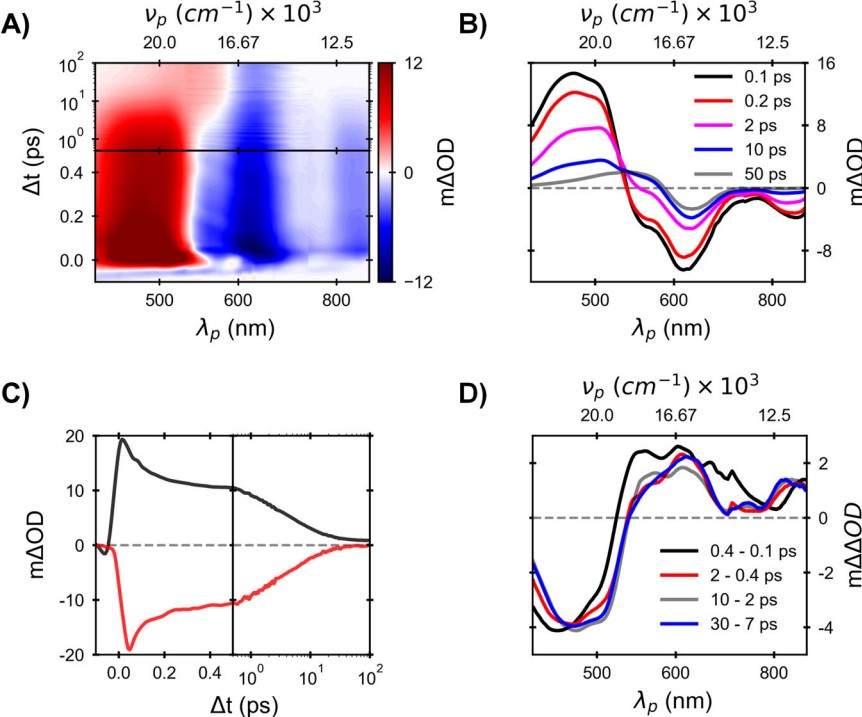

**Fig. 6 | Transient spectral data for K. A** The 2D color map of TA data of pure K, with X-axis representing probe wavelength or wavenumber and Y-axis representing the delay between pump and probe. Initial 500 fs pump-probe delay is linear, and the rest of the axis is logarithmic. The absorption difference is color-coded. The color bar at the right of the map presents the value of absorption difference. **B** TA Spectra at various pump-probe delay for K. **C** Excited-state absorption (ESA, 480 nm) and stimulated emission (SE, 870 nm) decay of K (SE decay is multiplied with 4). **D** Dynamic difference spectra at various stages of excited state's decay of K. Source data are provided as a Source Data file.

decay phases of the fluorescent state originating from an identical reactant and having the same photochemical outcome.

To shed further light on the multi-phased and ultimately slow excited state decay of K, the same QM/MM model was used to map out the relevant MEP. This required consideration of various possible conformations of the ASP212 sidechain for the relaxed K reactant. In particular, starting from the crystal structure of K[58], one could add a proton to either of the two oxygens of the ASP212 carboxylate group, namely O1 and O2, leading to the *syn* (s) or *anti* (a) conformations for each oxygen. These conformers, experimentally observed in solution[59] and known to play an important role in catalytic properties of enzymes[60], give rise to four initial models (OD1s, OD1a, OD2s and OD2a) with protonated ASP212. Upon ground state optimization only three of these survive, as the OD2a model collapses into OD2s. As shown in Fig. 7B–D, these three conformations of the complex counterion in K have similar energies at the CASPT2//CASSCF/MM level of theory, the most stable OD1s being just 2.7 kcal mol$^{-1}$ below OD1a, with OD2s situated in between. This is unlike AT and 13C where one dominant and stable conformer was found ((see Supplementary Table S2 in the Supplementary Information). This similarity in stability is consistent with the conservation of the total number of H-bonds in the counterion, i.e. five H-bonds spanning from ASP212 to w402, for them all. Notably, the ground state geometry optimization of the OD1a conformation yielded a planarization of the PSB's polyene chain (see Fig. 7D), suggesting that both pre-twisted and planar retinal structures are possible, as recorded in different crystal structures of the K intermediate[59,61].

Interestingly, the local electrostatic changes associated to these different conformations of the carboxylic acid group of protonated ASP212 appear to have a significant impact on calculated C=C bond relaxation MEPs (Fig. 7). In particular, for the most stable OD1s conformation, the energy profile is almost barrierless while increasing

energy barriers of 3.0 and 6.5 kcal mol$^{-1}$ are found for OD2s and OD1a respectively. Moreover, as shown in the Supplementary Information (see Supplementary Fig. 16), the C$_{13}$=C$_{14}$ photoisomerization computed MEPs for these three conformation feature barrierless) pathways towards the CI for OD1s and OD2s, and a 4.3 kcal mol$^{-1}$ barrier for OD1a, indicating the C=C bond relaxation path are the rate-determining step also for the excited state decay in K. Importantly, the standard deprotonation model features symmetric carboxylate groups that cannot exploit a variety of electrostatic local effects as protonated aspartates, excluding an interpretation of the multi-exponential decay from time-resolved experimental data as consequence of such electrostatic modulation. Furthermore, due to the specific location with respect to the PSB, the conformations associated with the protonation of the ASP85 sidechain have no effect on the C=C bond relaxation MEPs, which is found to be always barrierless (see Supplementary Fig. 17 in the Supplementary Information). This outcome further supports a theoretical model with protonated ASP212 for all resting states of BR, consistently with absorption linear and time-resolved experimental spectroscopies.

## Discussion

As conveyed in the introduction, the spectroscopy and function of BR should be critically influenced by hydrogen bond arrangement and protonation status of the counterion components in its various reactive states. The failure of the widely accepted structure for this crucial construct to reproduce these aspects of our findings has forced us to consider alternatives. Adopting a singly protonated assembly of the counterion appears remarkably capable of correcting much of that failure. Given the novelty of this suggestion we pause first to reflect on the suitability of the employed methods. Broad-band pump-probe experiments are often analyzed with global kinetic analysis in order to appreciate the underlying reactive mechanisms. The alternative

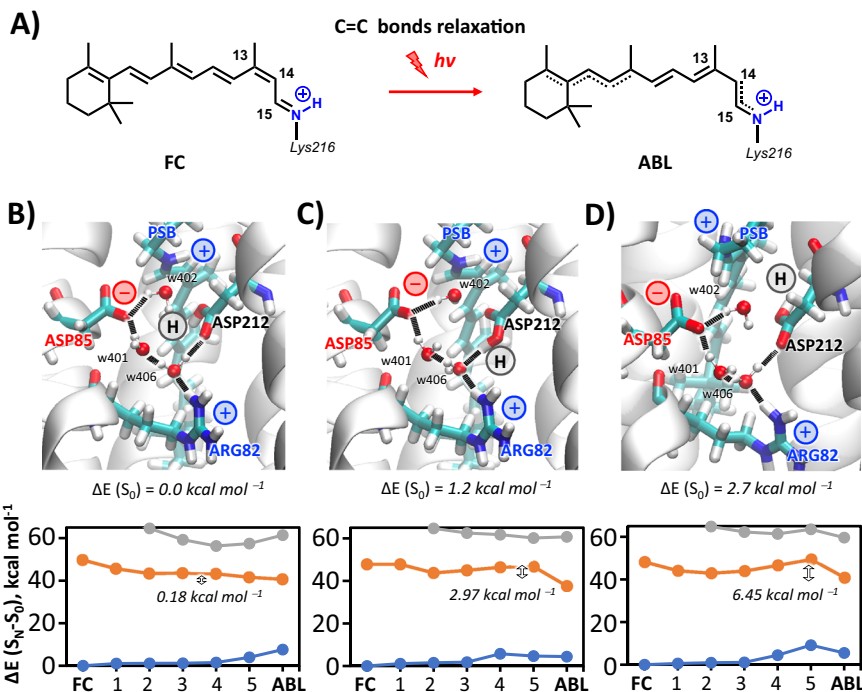

**Fig. 7 | The multi-exponential decay of the K intermediate. A** The C=C bond relaxation pathway from Franck-Condon (FC) to alternate bond-lengths (ABL) in the 13-*cis*, 15-*anti* K intermediate of AT-BR (all-*trans* Bacteriorhodopsin). The CASPT2//CASSCF/MM minimum energy paths (MEPs) of the FC→ABL in three different conformations of the carboxylic acid group of protonated ASP212, i.e., **B** OD1s, *syn* conformation-O1 protonated, **C** OD2s, *syn* conformation-O2 protonated and **D** OD1a, *anti* conformation-O1 protonated. The values 0.18, 2.97 and 6.45 kcal mol$^{-1}$ are associated with the energy barriers along the C=C bond relaxation path of OD1s, OD2s, and OD1a, respectively. Source data are provided as a Source Data file.

approach used here, which is based on impulsive Raman signatures and amplitude of stimulated emission in the NIR as unique markers for ground and excited state populations, is based on insights from previous photochemical studies of this and other MRPs in the Ruhman lab. These observables become available by extension of probing above 900 nm, which is the only range where bands from the reactive S$_1$ appear exclusively. Likewise, the ground state coherences are available through high S/N probing and sufficient time resolution for their detection. Time and again these have proven reliable for selectively detecting these states despite the extensive spectral overlap of signatures of ground, excited, and product states. These measures also are free of limitations introduced by modeling with kinetic schemes involving a limited number of distinct species in describing dynamic and continuous molecular evolution. This approach has been assessed by comparison to kinetic analysis and proven effective and often superior in dissecting electronic state evolution in the sample. Along with consistency tests described in the Supplementary Information we feel that this approach is reliable and capable of providing the sought-after information.

Likewise, it is of essence to consider the appropriateness of calculation methods employed, and to comment on reliability of calculated potential MEPs as a predictors of isomerization kinetics in this and associated proteins. CASPT2//CASSCF approaches within a hybrid QM/MM electrostatic embedding scheme have already shown their ability to quantitatively predict both steady-state and transient spectroscopic properties of animal[37,47,48,62,63] and archeal[47,64,65] rhodopsins, including artificial rhodopsin mimics[40,66]. These include decay rates and subtle features of their photoinduced dynamics such as oscillatory spectral modulation due to coherent wave packet motions[67]. In fact, absorption energies for a full set of retinal proteins and their mutants, spanning a 200 nm window from ca. 400 nm to ca. 600 nm, have been correctly reproduced with errors typically lying <0.2 eV (i.e., well below the ≈1.0 eV deviation found when using the standard BR

model)[47,48]. Remarkably, in this study BR appears as an outlier when employing the standard quadrupole model for the counterion. Interestingly, in a more recent work, BR absorption energy and ultrafast photoisomerization feature was recovered by protonation of one aspartate, although ASP85 was protonated in this case[68], strengthening the idea of a different electrostatic environment with respect to the standard counterion model. Finally, the very same CASPT2//CASSCF/MM MEPs based methodology has been used to predict and explain the behavior of far red shifted rhodopsins[69,70], including Neorhodopsin, a recently discovered MRP displaying the strongest red shifted absorption (690 nm) with intense fluorescence and no photoisomerization ability, thus showing that the employed approach may be successfully applied to predict the behavior of very different retinal proteins spanning diverse functions and absorption energies. Consistently, and confidently, in this work we extend it to three different resting states of BR, including the first stable photocycle intermediate K. Eventually, we show that unconventional protonation of ASP212 is the only arrangement of the counterion able to deliver consistent predictions of experiments, including both static properties and photoinduced dynamics.

After examining the chosen methods, the proposition that multiphased excited state decay is due to inhomogeneity in the configuration of a protonated ASP212 must be critically examined a) with respect to literature assertions that ASP212 is unprotonated in BR and relaxed K590, and b) vs all aspects of the pump-probe experiments described. Starting with the former we note that on the basis of 13C solid state NMR data[21,22] and FTIR spectroscopy of BR with ASP212 specifically labeled by 13C[71] it was proposed that ASP212 is deprotonated in the ground state. Furthermore, a study with BR mutants also suggested that ASP212 is deprotonated in BR ground state, and it is protonated only below pH 2 in which it is substituted by exogenous anions[72,73]. The low pKa of ASP212 was supported by electrostatic calculations[74,75]. FTIR spectroscopy suggested also that ASP212 remains unprotonated in the

M and N photocycle intermediates[76]. FTIR studies and theoretical calculations suggested that ASP212 is transiently protonated in the late photocycle O intermediate[77,78], indicating that ASP212 should be also deprotonated in the $K_{590}$ intermediate. Still, it is worth noting that computational studies supporting this view have never really considered the possibility of a protonated ASP212 as a valid starting hypothesis, relying on the interpretation of the experimental studies mentioned above. However, the present computational studies have shown that the FTIR results in the 2000–4000 cm$^{-1}$ region are not prone to unique interpretations. This holds also for mutations experiments, as already shown[79]. PKa calculations does also provide a typical example, as they heavily rely on structural data. These suffers from >1 Å uncertainties in atomic positions even for the better resolved X-ray protein structures whose electronic density maps are refined according to (and biased by) predefined models and that, besides H atoms, cannot resolve oxygens on very mobile waters as well as.

Furthermore, mutagenesis studies[49,80] revealed different effects on the photophysical properties of BR by mutating ASP85 or ASP212 with the same neutral residue (ASN). Notably, D212N exhibits slower photoisomerization dynamics compared to the wild-type (WT), in absence of a change in absorption maximum (being 568 and 566 nm in the WT and D212N, respectively), while D85N displays more pronounced slowdown of the photoisomerization dynamics compared to D212N and it is associated with a significant change of absorption maximum (largely red-shifted to 604 nm). This distinction is crucial since the large red-shift in D85N is consistent with a charge redistribution, i.e., with a deprotonated Asp85 in WT, but the very minimal change in absorption maximum in D212N mutant suggests a neutral ASP212 in WT, in contrast with the standard protonation model and in line with our model. Very notably, our model provides also a plausible explanation for the changes in photoisomerization dynamics resulting from point mutations, even when they do not correlate directly with shifts in absorption maximum, as experimentally observed for D212N. This explanation stems from the mutation's potential to influence the hydrogen bonding network (HBN) between the residue in question and neighboring amino acids. Our research underscores the substantial impact of HBN alterations on BR photoisomerization dynamics, a phenomenon we have elucidated across different protomers in the K-intermediate state.

Our model also provides a new perspective on the binding of Cl$^{-}$, observed upon D212N mutation[49]. Cl$^{-}$ is proposed to bind through an interaction involving ARG82, a component of the complex counterion. ARG82 is H-bonded to w406, which is in turn H-bonded to ASP212 in WT. Considering this, it's reasonable to anticipate that the D212N mutation might impact this HBN: ASN212 has an additional H-bonding donor feature (an extra H), while lacking the H-bonding acceptor feature due to the absence of a lone pair, compared to the OH group of protonated ASP212. Consequently, it is not surprising that Cl$^{-}$ binding affinity between D212N and the WT are different[49]. Moreover, it has been shown that ASP85 neutralization at low pH is key for Cl$^{-}$ binding[73], an observation that aligns with our model, which supports a low pKa for ASP85, indicating a deprotonated ASP85 in the ground state of BR, at odd with what recently suggested by an alternative computational model[47,68]. The possibility that ASP212 experiences deprotonation once ASP85 is protonated at low pH and that ASP212 is re-protonated at a very low pH accompanied by Cl$^{-}$ binding cannot be excluded.

Finally, it is worth noting that an absorption red shift and accompanying decrease on the photoisomerization rate is observed in BR upon protonation of ASP85 at low pH[81]. Our study scrutinizes the complementary effects due to different protonation states of ASP212, showing how such decrease on the photoisomerization ability and rate can be obtained also by deprotonating ASP212 and thus blue shifting the BR absorption energy. In both cases, i.e., protonated ASP85/ ASP212 and deprotonated ASP212/ASP85, the alteration of the photophysical properties is caused by the mixed covalent/ionic character displayed by the $S_1$ state, due to the stronger coupling with the covalent $S_0$ or $S_2$ states coming closer to $S_1$ when red or blue shifting the absorption, respectively, as shown in Fig. 8. Through the very numerous studies dedicated at exploring the photochemistry of microbial rhodopsins[47,70,82,83], which have significantly contributed to advancing our understanding, our work combined with that of Chang et al.[81] demonstrate that the new proposed counterion model creates the right electrostatic environment and HBN in BR in order to have $S_1$ exactly where it should be, i.e., far enough from both $S_2$ and $S_0$ covalent states to avoid mixing and to ensure a strong ionic $S_1$ character (Fig. 8A), which defines the electronic pre-requisite for optimal (i.e., ultrafast and efficient) retinal photoisomerization[17–20,44,65,84]. Thus, while our experimental and computational data primarily focus on BR, the insights can be extended well beyond it, eventually delivering a generalized energetic tuning mechanism (see Fig. 8) that accounts for the effect of the electrostatic embedding on both the spectral and photoisomerization properties of retinal chromophores in whatever environment (proteins, solvents or isolated condition).

Turning to b), the general suggestion that multiexponential photoisomerization dynamics in MRPs is often assignable to multiple protonation states of the reactants has been put forward in the past, including a recent paper by Tahara and coworkers[81]. The proposition here is more intricate since it also includes single bond rotamers and intra-carboxylate proton switching of the same formal protonation

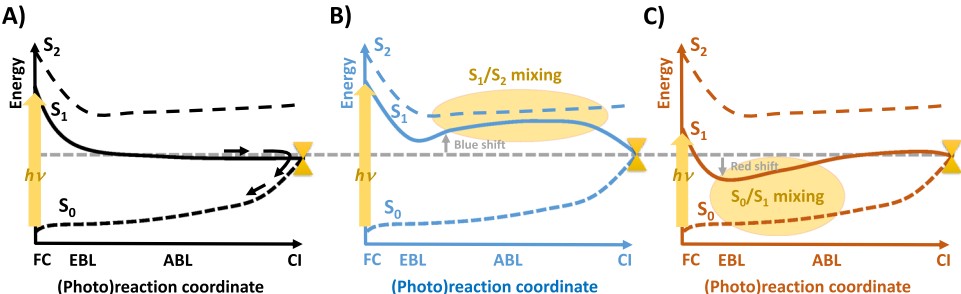

**Fig. 8 | General scheme for *electrostatic-induced* spectral tuning and excited state decay control in retinal systems. A** Well-separated $S_1/S_2$ and $S_0/S_1$ energy gaps eliminates electronic mixing-induced barriers along the $S_1$ decay path, thus facilitating a fast and efficient decay and photoisomerization (this is what happens in the new BR counterion model presented here, Rhodopsin[18,37,48,84] and the gas phase[18]). **B** Blue shift and strong $S_1/S_2$ (ionic/covalent) mixing (as it happens in solvents[17] or in BR when adopting the standard quadrupole counterion model) resulting in a $S_1/S_2$ mixing-induced barrier along the photoisomerization paths that slows down decay. **C** Red shift and strong $S_1/S_0$ mixing (BR at low pH, giving a protonated ASP85)[81], resulting in a $S_0/S_1$ mixing-induced barrier also slowing down decay and photoisomerization (the heavily red shifted, to 690 nm, Neorhodopsin being the extreme case, showing intense fluorescence with no photoisomerization)[70]. ABL and EBL points refer to alternate bond lengths and equalized bond lengths, respectively.

state. In terms of the finite difference spectra characteristic of the latter two reaction phases shown in Fig. 6D, their identical appearance indicates either that the reactants, products and quantum efficiencies of both are identical, or that despite the distinct nature of the reactants all these characteristics are coincidentally identical except for the isomerization rates. Explaining this coincidence will require further study. The structure and spectral width of K absorption spectrum determined here and in earlier studies might exhibit signs of inhomogeneous broadening due to the suggested distribution of initial structures. Indeed, the expected trend is right since spectra extracted for the first ground state photo-cycle intermediate tends to be 10–15% broader than that of the AT resting state $BR_{568}$ for instance.

In conclusion, despite expectations of extremely rapid photoisomerization of "K" based on pre-twisting of its retinal chromophore, our experiments show it is multi-phased, with an average decay time far exceeding that observed for either the AT or 13C resting states. Data analysis further unexpectedly demonstrated that a majority of BR in the purple membrane remains in the AT configuration even after equilibration in the dark, rendering it biologically viable even in the first light of day. To interpret the dependence of photo-dynamics on the protein's initial state an alternative counterion model has been introduced that challenges the conventionally accepted quadrupole in BR. Our findings demonstrate a good fit to many previously published data while improving spectral fitting and explaining the observed differences in photoisomerization dynamics. Further study including structural and mechanistic investigations of the different BR resting states (DA, LA and K intermediate) and binding of $Cl^-$ at a very low pH value will be required for validation of this new counterion protonation scheme. Overall, our model calls for a reevaluation of previous experimental findings, which should be approached with a critical eye in light of this alternative perspective, raising important questions about the electrostatic environment and protonation states in microbial rhodopsins.

## Methods

### Sample preparation
Wild-type bacteriorhodopsin was isolated from *Halobacterium salinarum* strain S9 as purple membranes as was previously described[85]. The grown cells were harvested by centrifugation using 7000 rpm for 15 min. The cells (about 50 gr.) were suspended in 250 mL Basal salt solution and were incubated with 5 mg Dnase for 1 h at 25 °C followed by dialysis against 0.l M NaCl. The suspension was centrifuged at 7000 rpm for 7 min and the precipitate was discarded. The suspension was further centrifuged using 16000–20000 rpm for 1–1.5 h, until the purple patches were precipitated. The pellet was suspended and centrifuged once again using similar conditions. This procedure was repeated for about 5–10 times, until the mother liquor is clear and the absorbance ratio of the PM suspension at 280 nm/560 nm was 2.

### Experimental setup
**Absorption and TA measurement.** Absorption and TA measurements are performed on a home build-follow cell containing a 300-micron quartz window and 200-micron path length. During a measurement, the sample was continuously flowing using a syringe pump. The flow rate was tuned so that individual laser pulsed met with a fresh sample. To light adapt the sample one of the reservoirs was illuminated with a 150 W halogen fiber light source. A low-pass filter is installed on the output of the fiber to avoid heating. To achieve dark adaptation sample was kept in the dark overnight. To keep the sample under dark adaptation for the measurements, 1.5 ml of the BR sample is used. Sample integrity, and light and dark adaptation are monitored consistently during the measurement.

**Visible probe TA.** Transient absorption (TA) measurements were carried out using a hybrid Ti-sapphire base multipass amplified laser

system. The seed pulse for the amplification was taken from an oscillator (Vitara, Coherent) that produced about 60 million 20 fs pulses center at 800 nm each second. Using a home-built amplifier setup, an oscillator pulse was stretched, amplified, and compressed every millisecond to obtain a 30 fs, 1 mJ pulse around 800 nm. The laser output was divided into three components 50:30:20. A high power output fraction was used to seed two second-harmonic pumped noncolinear optical parametric amplification (NOPA) setups to generate tunable narrow or broadband pulses in the 500–700 nm range. Broadband NOPA pulses were compressed using a chirp mirror pair (DCM 12, Laser quantum). Broadband pump pulse compression was characterized using a Kerrgated frequency resolved optical grating (FROG) method on a 150-micron quartz medium with a fundamental gate pulse[86]. Medium power fundamental fraction pumps a colinear optical parametric amplifier (Topas 800, Light conversion) that produces a tunable pulse in NIR. The supercontinuum probe pulse in the 440 to 880 nm wavelength range was generated by focusing about 2 mJ 1200 nm OPA output into a 2 mm thick calcium fluoride window. $CaF2$ was mounted on a motorized continuous rotation mount to prevent laser damage. Pump and probe beams were focused in the sample using 20 and 15-cm focal-length spherical mirrors, respectively. The transmitted probe beam is dispersed through an imaging spectrograph (Spectrapro 2150i, Actron Research Corporation) and detected on a Si-array CCD detector (Entwicklungsbüro Stresing). A mechanical chopper (MC1000, Thorlabs) that ran at 500 Hz was fitted on the pump arm to modulate the pump. The shot-to-shot absorption difference (DOD) is calculated between the pump-on and pump-off measurements. A computer-controlled motorized translation stage (Physik Instrumente) was fitted on the probe arm to control the delay between the pump and probe beams precisely. TA data set includes DOD measurements for a series of pump-probe delays.

Polarization of the pump beam was rotated with respect to the probe beam using a broad band achromatic half-wave plate. A broadband polarizer was placed on the probe before the spectrograph to prevent pump scattering. Two TA measurements are performed for each experiment, one with identical polarization and another with orthogonal pump-probe polarization.

An additional pump pulse was taken from the narrow band NOPA for the three-pulse TA measurement. The actinic pulse was translated through a motorized delay stage (Physik Instrumente). TA measurements were performed 60 ps after the actinic excitation.

### NIR probe TA
NIR TA measurements in the 900–1400 nm range were performed using the same laser system at 370 Hz repetition rate. A supercontinuum probe pulse was generated by focusing about 2-mJ fundamental on a 3 mm thick static sapphire window and detected on an InGaAs array detector (B&W Tek).

### Vibronic modulation
The initial part of the TA data, up to 80 fs, contains complex signatures due to the Kerr effect and finite width of the pulses. The rest of the TA data contains population kinetics and coherent modulation information. Oscillating coherent modulations are isolated from the TA data by subtracting the biexponential fit of the kinetics.

### Computational details
**QM/MM model construction.** The x-ray crystal structure of BR in different states (AT; 5ZIL, 13C; 1X0S, and K; 1M0K) was used for geometry optimization and models construction. Considering two ASP (ASP85 and ASP212) which participate in the complex counterion each with two oxygen of carboxylate group, five different protonation states are possible: 1) Both ASP are deprotonated; corresponds to standard complex counterion protonation states). ASP85 is protonated either on

2) OD2 or on 3) OD1 oxygen; denoted as ASH85-OD2 and ASH85-OD1, respectively. (Oxygen atom types named based on the amber convention to distinct two oxygens of carboxylate group). ASP212 is protonated either on 4) OD2 or on 5) OD1 oxygen; denoted as ASH212-OD2 and ASH212-OD1, respectively. Two different protocols applied for model construction and geometry optimization of five protonation states of three BR states, as follow:

Protocol-1: In this model, the QM part consists of the rPSB (starting from the Cε of the Lys216), sidechains of the ASP85, ASP212 and ARG82 and three crystallographic waters participating in the H-bond network of the quadrupole in the standard protonation state (Supplementary Fig. 1). The MM part consists of the main chain of amino acids ASP85, ASP212, ARG82 and the Lys216. The amino acids ASP85, ASP212 and ARG82 were cut between Cα and Cβ carbon while the covalently linked Lys216 to the rPSB11 chromophore was cut after Cε, whereas a hydrogen link atom was used to saturate the QM region. The remaining part of the system (including protein and crystallographic waters) were kept frozen in the low layer in their x-ray position. All QM/MM calculations were performed with the COBRAMM package[87] with electrostatic embedding, interfaced with Amber (MM)[88] and Gaussian 16 (QM)[89]. Ground-state optimizations were performed at density functional theory (DFT) level using the B3LYP functional[90,91] and the 6−31 G(d) basis set.

Protocol-2: Starting from the models obtained above, the environment was kept frozen in its optimized DFT position and the geometry of rPSB was further optimized at the CASSCF/MM level with an (12e,12o) active space that encompassed the whole π-system of rPSB11. MOLCAS 8.0[92] and Amber[88] were used for QM/MM computations via the COBRAMM package[87]. To overcome the problem of intruder states, the so-called "imaginary shift" method has been used as implemented in Molcas 8.0, with a value of 0.2 a.u., and the zero-order Hamiltonian shift (IPEA shift) set to zero.

Throughout this work electronic (LA) and vibrational (IR) spectroscopy has been used to validate the constructed models of three BR states and assign the correct protonation state of the quadrupole in AT state, respectively. Accordingly, the excited-state electronic structure (i.e. vertical excitation), normal modes, and associated frequencies have been computed on top of the optimized geometries of all interested systems. Further details have been provided in Supplementary Note 1.

**Semi-classical excited state dynamics.** Franck-Condon (FC) trajectories are semi-classical (i.e. non-adiabatic) trajectory starts at the equilibrium structure of the QM/MM models (obtained from part 2) with zero initial velocities. The ultrafast lifetime of the BR photoisomerization ensure that these trajectories describe the average evolution of the corresponding excited state population. Since we are only interested in the evolution from the FC point to the decay to $S_0$ (i.e. in the region of $S_2/S_0$ and $S_1/S_0$ conical intersection in deprotonated and protonated states, respectively), all trajectories were propagated at CASSCF/6−31G(d)/MM level of theory, according to a deterministic surface-hop method with a 0.5 fs time step, as implemented in the COBRAMM package[87].

**Minimum energy path (MEP).** Characterization of the MEPs executed for the two main relaxation coordinates involved in rPSB photoisomerization. First, bond length relaxation coordinate has been studied by single point scans (with frozen bond lengths) at CASPT2/6−31G(d)/MM level of theory, between the FC and the ABL (alternated bond length) geometries. The initial coordinates of the intermediate structures acquired by linear interpolation of those single-double bonds alternate during the photo-isomerization process. The geometry of ABL point was obtained by optimization of the first ($S_1$) excited state. Second, characterization of the MEPs associated with the isomerization around the central $C_{13}$−$C_{14}$ bond was performed,

starting from the ABL structure, with gradual rotation of the $C_{12}$−$C_{13}$=$C_{14}$−$C_{15}$ dihedral angle by 14 degrees (with the atom numbering given at Fig. 3). The optimization scan involved constraints for central dihedral at CASSCF/6−31G(d)/MM, followed by the energy correction at CASPT2/6−31G(d)/MM level of theory.

## Data availability
The CASSCF/MM geometries for the systems light-adapted, dark-adapted and K intermediate BR used for the computation of bond relaxation and dihedral photoisomerization minimum energy pathways (MEPs) in Fig. 3 and Fig. 7, have been deposited in the Zenodo database under the accession code 10472281[93] (https://doi.org/10.5281/zenodo.10472281). Source data are provided with this paper and the Supplementary Information. Source data are provided with this paper.

## Code availability
The COBRAMM software[87] suite employed to produce QM/MM simulation results is distributed under a GNU General Public License v3.0 (GPL v3) and can be obtained as a free of charge open source code from the public COBRAMM GitLab repository at the following link: https://gitlab.com/cobrammgroup/cobramm

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

## Acknowledgements

S.R. holds the Lester Aronberg Chair in Chemistry. S.R. thanks the US Israel binational science foundation for grants 2016102 and 2020105. I.R. acknowledges funding from the Italian Ministry of Education, University and Research (MIUR) program PRIN 2020, project PSI-MOVIE, prot. 2020HTSXMA. M.G. Acknowledges the Italian Ministry of Education, University and Research (MIUR) program PRIN 2017, project PHANTOMS, prot. 2017A4XRCA.

## Author contributions

P.M. handled and conducted experimental measurements. S.G. and M.A. carried out the computations. M.G., I.R., S.R., and M.S. designed and supervised the project. I.R., S.R., M.G., and S.G. contributed to drafting the initial paper. P.M., S.G., M.S., and M.A. analyzed the data. I.R., P.M., S.G., and M.A. contributed to providing figures.

## Competing interests

The authors declare no competing interests.
