## [Peer Review File · Nature Communications]

Editorial Note: Parts of this peer review file have been redacted as indicated to maintain the confidentiality of other journals.

REVIEWERS' COMMENTS

Reviewer #1 (Remarks to the Author):

I am reviewer 1 in the previous review of [redacted] manuscripts. The authors have addressed all my comments in the revised manuscript. I recommend the publication of this manuscript.

Reviewer #3 (Remarks to the Author):

In the first iteration of the review process several issues were pointed out which have been – to a large degree – addressed accordingly by the authors. The revised manuscript now includes suggested changes (e.g. additional conceptual explanations, additional graphical material), which significantly improves the readability and the appeal for specialists in rhodopsin research, as well as a broader audience of physical chemists and photochemists. However, there are still a few minor points which should be addressed/considered for the final manuscript:

1.) Taken from the response letter:

“While our manuscript primarily focuses on BR, this scheme has general validity beyond BR. Regarding the recent study done by Chang et al. (Angewandte Chemie, 2021), this important work was already cited (ref. 80) in our original manuscript and now it is used as starting point in the revised manuscript to further support and elaborate on our findings.”

And the respective added paragraph “... Very remarkably, the work of Chang et al. 80 and our study taken together show ...”

Even though Chang et al. proposed a unified view on the retinal photochemistry in microbial rhodopsins, many other groups in the community contributed as well to this development. Those contributions should not be shrouded under such a statement and should be at least acknowledged accordingly. Albeit it is understandable that not every publication can be considered and cited, this narrow discussion does not do justice to the current state of research.

2.) Minor suggestions to improve the figures:

Figure 4: The (colours chosen for the) calculated spectra are difficult to distinguish, especially in the range of 2.1 eV where they overlap with the experimental spectra.

The multiplication symbols “.” behind the L (in Fig.4) and again for the unit of the extinction coefficient (y-axis of figure 5A) are strange.

Figure 5B: The colours of the indicated pulses (above the arrows) have very low contrast compared to a white background

Figure 8: The meaning of EBL and ABL should be reiterated in the figure caption (similar to the caption of Figure 3)

Figure S9: The labels of the residues, as well as the labels of the IR spectra are difficult to read. Especially for the IR spectra in S9E and S9G it is recommended to use the available space more efficiently and enlarge the spectral features.

3.) Please check the consistent use of “-”, e.g. DA-BR vs. LA BR (page 7 of the revised manuscript) or 13C BR vs. 13C-BR (also page 7 in the caption of Figure 2)

Reviewer #3 (Remarks to the Author):

In the first iteration of the review process several issues were pointed out which have been – to a large degree – addressed accordingly by the authors. The revised manuscript now includes suggested changes (e.g. additional conceptual explanations, additional graphical material), which significantly improves the readability and the appeal for specialists in rhodopsin research, as well as a broader audience of physical chemists and photochemists. However, there are still a few minor points which should be addressed/considered for the final manuscript:

1.) Taken from the response letter:

“While our manuscript primarily focuses on BR, this scheme has general validity beyond BR. Regarding the recent study done by Chang et al. (Angewandte Chemie, 2021), this important work was already cited (ref. 80) in our original manuscript and now it is used as starting point in the revised manuscript to further support and elaborate on our findings.”

And the respective added paragraph “... Very remarkably, the work of Chang et al. 80 and our study taken together show ...”

Even though Chang et al. proposed a unified view on the retinal photochemistry in microbial rhodopsins, many other groups in the community contributed as well to this development. Those contributions should not be shrouded under such a statement and should be at least acknowledged accordingly. Albeit it is understandable that not every publication can be considered and cited, this narrow discussion does not do justice to the current state of research.

We appreciate the reviewer's valuable comments and acknowledge the significance of recognizing the contributions of various research groups in the photochemistry of microbial rhodopsins. In the revised manuscript, we have expanded our acknowledgment of the broader research community involved in understanding microbial rhodopsins, building upon the foundational work of Chang et al. (Angewandte Chemie, 2021), which we cited as a starting point to support and elaborate on our findings.

We have accordingly revised the main text of the manuscript as following: “Through the very numerous studies dedicated at exploring the photochemistry of microbial rhodopsins^{47,70,82,43} which have significantly contributed to advancing our understanding, our work combined with that of Chang et al.⁸¹ demonstrate that the ...”

2.) Minor suggestions to improve the figures:

Figure 4: The (colours chosen for the) calculated spectra are difficult to distinguish, especially in the range of 2.1 eV where they overlap with the experimental spectra.

We value the reviewer's comment related to the Figure 4 and confirm that the concern has been addressed. The corresponding changes have been made, and the colors adjusted accordingly.

The multiplication symbols “.” behind the L (in Fig.4) and again for the unit of the extinction coefficient (y-axis of figure 5A) are strange.

In response to this feedback, we have revised the figure to present the symbols in a more standard and clear manner. The updated figure ensures improved visual clarity and alignment with common conventions.

Figure 5B: The colours of the indicated pulses (above the arrows) have very low contrast compared to a white background

Thank you for your valuable feedback regarding Figure 5B. We have addressed this concern by modifying the colors of the indicated pulses in accordance with your suggestion, aiming to enhance the contrast against the white background. The adjustments made should improve the overall visibility and readability of the figure.

Figure 8: The meaning of EBL and ABL should be reiterated in the figure caption (similar to the caption of Figure 3)

We have revised the figure caption to include an explanation of the abbreviations EBL and ABL, similar to the caption of Figure 3. Thanks for the discerning observation.

Figure S9: The labels of the residues, as well as the labels of the IR spectra are difficult to read. Especially for the IR spectra in S9E and S9G it is recommended to use the available space more efficiently and enlarge the spectral features.

Thank you for the valuable feedback regarding Figure S9 (in the current version: Supplementary Figure 10). We have taken your comments into consideration and made the necessary adjustments to improve the legibility of both the residue labels and the IR spectra labels. Specifically, we have enhanced the utilization of available space and enlarged the spectral features in Figures S9E and S9G (Supplementary Figure 10E and 10G in the current version of the SI, respectively) to ensure better readability.

3.) Please check the consistent use of “-“, e.g. DA-BR vs. LA BR (page 7 of the revised manuscript) or 13C BR vs. 13C-BR (also page 7 in the caption of Figure 2)

We have thoroughly reviewed the document, specifically on page 7 as indicated, and ensured the consistent application of hyphens in terms such as DA-BR and LA-BR. Additionally, we have addressed the inconsistency noted in the caption of Figure 2, correcting it to maintain uniformity (e.g., 13C-BR).